# The Emerging Physiological Role of AGMO 10 Years after Its Gene Identification

**DOI:** 10.3390/life11020088

**Published:** 2021-01-26

**Authors:** Sabrina Sailer, Markus A. Keller, Ernst R. Werner, Katrin Watschinger

**Affiliations:** 1Biocenter, Institute of Biological Chemistry, Medical University of Innsbruck, 6020 Innsbruck, Austria; sabrina.sailer@i-med.ac.at (S.S.); ernst.r.werner@i-med.ac.at (E.R.W.); 2Institute of Human Genetics, Medical University of Innsbruck, 6020 Innsbruck, Austria; markus.keller@i-med.ac.at

**Keywords:** AGMO, tetrahydrobiopterin, alkylglycerols, plasmalogens, neurodevelopment, autism, inflammation, macrophages, type 2 diabetes, energy homeostasis

## Abstract

The gene encoding alkylglycerol monooxygenase (AGMO) was assigned 10 years ago. So far, AGMO is the only known enzyme capable of catalysing the breakdown of alkylglycerols and lyso-alkylglycerophospholipids. With the knowledge of the genetic information, it was possible to relate a potential contribution for mutations in the AGMO locus to human diseases by genome-wide association studies. A possible role for AGMO was implicated by genetic analyses in a variety of human pathologies such as type 2 diabetes, neurodevelopmental disorders, cancer, and immune defence. Deficient catabolism of stored lipids carrying an alkyl bond by an absence of AGMO was shown to impact on the overall lipid composition also outside the ether lipid pool. This review focuses on the current evidence of AGMO in human diseases and summarises experimental evidence for its role in immunity, energy homeostasis, and development in humans and several model organisms. With the progress in lipidomics platform and genetic identification of enzymes involved in ether lipid metabolism such as AGMO, it is now possible to study the consequence of gene ablation on the global lipid pool and further on certain signalling cascades in a variety of model organisms in more detail.

## 1. Introduction

In 1964, Tietz, Lindberg, and Kennedy were the first to describe a novel enzymatic reaction system requiring tetrahydrobiopterin and molecular oxygen for the catabolism of the 1-O-alkyl bond in ether lipids [1]. Since then, several groups have been investigating the biochemistry behind this system, calling it glyceryl-ether monooxygenase (EC 1.14.16.5), which was later on referred to as alkylglycerol monooxygenase (AGMO) according to the enzymatic reaction in which it is involved [2,3,4,5,6,7,8]. However, for more than four decades, this ether lipid-cleaving enzyme was considered an orphan enzyme, which is a protein with known catalytic function but unknown genetic information, making it difficult to study its physiological role in more detail. Classical standard approaches to isolate and sequence the AGMO protein were not applicable, because it is a highly hydrophobic integral membrane protein that is very labile to solubilisation and purification. Finally, in 2010, Watschinger and colleagues were able to assign the gene TMEM195 to AGMO [9].

### 1.1. Ether Lipids and Their Biosynthesis

Glycerol-based ether lipids characteristically carry a side chain attached to the sn-1 position by either an alkyl-bond (alkylglycerols or plasmanyl phospholipids) or a vinyl-bond (plasmalogens or plasmenyl phospholipids). Plasmalogens make up about 20% of the total phospholipid pool in humans [10]. The biosynthesis of ether lipids starts in peroxisomes from glycerone phosphate (previously called dihydroxyacetone phosphate (DHAP, ①)), which is processed by glycerone-phosphate O-acyltransferase (I, EC 2.3.1.42), alkylglycerone phosphate synthase (III, EC 2.5.1.26), and acyl/alkyl-glycerone phosphate reductase (IV, EC 1.1.1.101) (encoded by the gene DHRS7B; also known as PexRap [11]) and is then completed in the endoplasmic reticulum (Figure 1). A crucial, rate-limiting step in the pathway is the provision of fatty alcohols ③ by fatty acyl-CoA reductase (II, EC 1.2.1.84). The pathway branches to plasmalogens after reaction VII, where an ethanolamine is attached instead of the choline (by ethanolamine-phosphotransferase, XIV, EC 2.7.8.1) to form plasmanylethanolamine ⑭, which is then desaturated by plasmanylethanolamine desaturase (XV, (EC 1.14.19.77, until 2020 EC 1.14.99.19) to form the characteristic vinyl ether bond of plasmalogens ⑮ [12,13].

### 1.2. The AGMO Reaction

So far, AGMO is the only enzyme known that is able to cleave the ether bond of free alkylglycerols ⑬ and lyso-alkylglycerophosphocholines ⑧ (but also lyso-alkylglycerophosphoethanolamines are accepted) (Figure 1). However, the substrate specificity of AGMO has to be interpreted with caution, because all studies so far were performed in homogenates and not with a pure AGMO enzyme. In 2017, a biosynthetic enzyme in *Pseudoalteromonas rubra* related to AGMO was discovered, which catalyses the cyclisation of prodigiosin to cycloprodigiosin [14].

For the enzymatic reaction, AGMO requires tetrahydrobiopterin and molecular oxygen to oxidise the alkyl bond at the sn-1 position, which generates a hemiacetal ⑨ that rearranges to a glycerol derivative ⑩ and a fatty aldehyde ⑪ that is toxic for the cells and therefore converted by fatty aldehyde dehydrogenase (XIII, EC 1.2.1.48) to the corresponding fatty acid ⑫ [15] (Figure 1).

Compared to ester lipids, the role of ether lipids is still not fully understood, since their detection has not been routinely included in standard lipid analysis protocols in the past decades. Glycerol-based ether lipids contain a fatty alcohol attached to the sn-1 position by either an alkyl- (plasmanyl ether lipids) or vinyl-bond (plasmenyl ether lipids, better known asplasmalogens) that confers distinct chemical, biochemical, and biophysical properties [11].

The identification of the sequence encoding for AGMO enabled further insights into its biology, localisation, chemistry, and role in ether lipid degradation during the last 10 years. AGMO is localised to the endoplasmic reticulum and shares a common fatty acid hydroxylase motif of which the characteristic eight conserved histidines are all crucial for the enzymatic activity [16]. In fact, also, a ninth histidine is conserved among fatty acid hydroxylases (Figure S7 in [9]), which also turned out to be essential for enzymatic activity [16] (Figure 2).

### 1.3. Structural Features of the AGMO Protein

All fatty hydroxylases known so far are integral membrane proteins, making them very labile in isolation and thus difficult to purify. Although Ishibashi and Imai described a protocol to purify and solubilise AGMO [17], all further attempts to reproduce purification failed, underlining the unstable nature of this protein. Using Rosetta Membrane *ab initio* modelling, a putative model of a part of the AGMO protein (amino acids 37 to 205) was generated, which contained five of the conserved eight histidines that shape the di-iron centre crucial for iron binding and catalytic activity. By hydropathy plots, nine transmembrane domains were proposed [16] (Figure 2).

A comprehensive review on AGMO substrate specificity, co-factor kinetics, inhibitors, and enzyme-based activity assays was published in 2013 [18]. This review here will focus on novel insights into the putative physiological role of AGMO acquired during the last 10 years since the discovery of its gene.

## 2. Alkylglycerols

### 2.1. Occurrence and Properties of Alkylglycerols

Alkylglycerols and alkylglycerophospholipids account for a considerable amount of the total lipid pool in single-celled organisms such as archaea and marine organisms but are less abundant in mammalian cells [19]. However, their precise physiological role is poorly understood. Due to the alkyl-bond that lacks the carbonyl oxygen at the sn-1 position, these side chains are resistant to acid treatment as compared to plasmalogens. Chemical properties such as the melting temperature, packing, and hydrophobicity of lipids with an alkyl side chain and membranes containing ether lipids may also vary. Nonetheless, the physicochemical nature is not exclusively influenced by the ether bond but may also depend on the headgroup such as in phospholipids, the degree of saturation and the length of the side chain underlining the complex behaviour of lipids [20]. In *C. elegans* AGMO mutants with an altered ether lipid profile, buoyancy is influenced [21], and in deep-sea sharks, diacyl glycerol ethers are essential components for the buoyant capacity of the liver as a swim bladder substitute [19]. However, the precise role of endogenous mammalian alkylglycerol metabolism, in which AGMO is involved, has been marginally investigated.

### 2.2. Dietary Supplementation of Alkylglycerols

In earlier studies (reviewed in [22]), dietary supplementation with alkylglycerols, such as shark liver oil where these lipid species are highly abundant, was shown to exert immunomodulatory stimuli by macrophage activation [23]. In mouse models, this conferred anticancer properties [24]. Alkylglycerols were shown to be able to open the blood brain barrier [25] and have antipathogenic capacities [26]. Alkylglycerol derivatives were shown to be also important second messengers and potent modulators of a variety of kinases including protein kinase B and C or phosphoinositide-3-kinase [10]. Supplementation studies with high doses of alkylglycerols showed beneficial effects on weight and insulin sensitivity in high-fat diet-induced obesity in mice [27]. In obese humans (body mass index ≥30 and ≤40 kg/m^2^), a clinical supplementation trial with alkylglycerols reduced serum complement levels and attenuated the prevalence for cardiovascular risk [28]. Yet, a more recent study identified alkylglycerols in breastmilk to be important for macrophage signal transducer and activator of transcription 3 (STAT3)-mediated maintenance of beige adipose tissue, thereby preventing conversion into white adipose tissue [29].

### 2.3. Interdependence of Ether Lipid and Sphingolipid Metabolism

Lately, another complex interplay between ether lipids and other lipid classes has been shown on the basis of a combinatorial approach using genome-wide CRISPRi library and inhibition of sphingolipid biosynthesis by the antibiotic myriocin [30]. A tight connection between ether lipid and sphingolipid synthesis in mammalia has been implicated in cellular survival. Furthermore, an inverse reciprocal interplay between ether phosphatidylcholine and ceramide levels was shown. In that study, ether phosphatidylcholine was postulated to support the anterograde p24-driven vesicular transport of glycosylphosphatidylinositol anchors, whereas sphingolipids exerted *vice versa* action and coordinated retrograde trafficking to the endoplasmic reticulum [30].

### 2.4. Analytics of Ether Lipids

With the progress in lipid analysis, alkyl diacyl glycerol ethers (referred to as monoalkyl diacylglycerols, MADAG) were found to be accumulated towards the completion of adipocyte differentiation in several *in vitro* models of adipocyte differentiation and also in adipose mouse tissues [31,32]. A role of AGMO in regulating the concentrations of these compounds remains to be established. Key to expanding our knowledge about the physiology and regulation of AGMO functions in the past ten years was the analytical tools that were either specifically developed for studying this enzyme, as well as further existing approaches that have been used for these purposes. An illustrative example for this is that with establishing the possibility to sensitively and directly determine the enzymatic activity of the AGMO using fluorescent lipid substrates combined with HPLC [33], it was ultimately possible to successfully assign the gene coding for this enzyme function [9].

### 2.5. Discrimination of 1-O-Alkyl- and 1-O-Alk-1′-Enyl-Lipids

For the investigation of consequences of the manipulation of AGMO activity in the lipid metabolic network, it is important to discriminate 1-O-alkyl-glycerol derivatives, which can be cleaved by AGMO when containing a free hydroxy group at sn-2, from 1-O-alk-1′-enyl lipids, such as plasmalogens, which are degraded by different enzymes. Liquid chromatography-tandem mass spectrometry (LC-MS/MS)-based techniques are the methods of choice also because the many lipid species are impossible to be separated and quantified using traditional methods such as thin-layer chromatography [34]. A fundamental problem that exists in relation to the measurement of lipids by means of mass spectrometry is the large number of isomeric species, which makes it very difficult to clearly identify molecular species. This especially applies to the distinction between plasmanyl and plasmenyl lipids, which are of particular importance in connection with research on AGMO [35]. One strategy to deal with such imprecise annotations is the use of lipid class enrichment techniques [36]. Different analytical strategies have been developed to properly discriminate different ether lipid subclasses, including the generation of structurally informative lithium-salt adducts [37] and the implementation of instrumentally more demanding MS^3^ and MS^4^-fragmentation experiments [38]. Recently, it has been systematically demonstrated that the chromatographic properties of plasmanyl and plasmenyl lipids are different enough to facilitate their univocal mapping in LC-MS/MS lipidomic experiments [39].

## 3. AGMO in Human Diseases

### 3.1. Genome-Wide Association Studies

As soon as the genetic information for AGMO was available, it was possible to link variants in the TMEM195 gene to AGMO in genetic analyses and get an idea about the physiological role in humans. Association studies identified a possible role in glucose homeostasis and the prevalence to develop type 2 diabetes [40,41,42]. Genome-wide association studies of the Meta-Analyses of Glucose and Insulin-Related Traits Consortium identified a single nucleotide polymorphism (rs2191349) lying in the intergenic region between diacylglycerol kinase beta (DGKB) and TMEM195/AGMO that was strongly associated with fasting glucose, fasting insulin, and indices of beta-cell function [40]. This finding was confirmed in the Inter99 cohort by oral glucose tolerance tests and also related single nucleotide polymorphism rs2191349 to decreased insulin response [41]. The same single nucleotide polymorphism was additionally associated for colorectal cancer in overweight women [43]. Later on, follow-up studies supported these results [44,45,46,47,48,49]. AGMO is highly expressed in liver, and a connection to energy homeostasis might be plausible but has not been experimentally proven yet. Thus, further analyses are required to reveal if AGMO function and ether lipid degradation connect to glucose metabolism. Other single nucleotide polymorphisms (rs4628172 and rs7781293) were associated to correlate with a predisposition to develop intracranial aneurysms in a Japanese case-control study [50].

From genome-wide association studies, an additional 31 further single nucleotide polymorphisms either in the AGMO locus or in the intergenic region between AC006458.1 or MEOX2 were associated with a variety of pathologies such as obesity and related disorders [51,52], diabetes and related disorders [53,54,55], tuberculosis [56], lung function [57,58,59], cancer [60,61], heel bone density [62,63], adolescent idiopathic scoliosis [64], metabolism [65], sweet taste preference [66], blood zinc levels [67], and neurological disorders [68,69,70,71,72,73]. However, further evidence needs to be collected, as most of these associations were found only in one study and might be of minor relevance.

### 3.2. Mutations, Copy Number Variations, and Deletions in the AGMO Gene

#### 3.2.1. Autism Spectrum Disorders

AGMO has also been implicated to play a role in neurodevelopmental disorders such as autism spectrum disorders. Already before the gene TMEM195 was assigned to AGMO, a spontaneous copy number variation in the region (then called FLJ16237) was identified, leading to a deletion within exons 2–8 in a patient with autism and was strongly correlated to autism spectrum disorders [74]. Three years later, *de novo* mutations within this gene were confirmed to be involved in autism in another cohort study [75]. Later, other groups confirmed a correlation of rare copy number variations and mutations within the AGMO locus with autism [76,77,78,79,80]. So far, 10 mutations or copy number variations are associated and nine variants are annotated, of which three occurred *de novo* and four are inherited in a familial pattern (http://autism.mindspec.org/GeneDetail/AGMO).

#### 3.2.2. Microcephaly

A compound-heterozygous deletion (c.967delA; p.Glu324Lysfs12*) on both alleles of the AGMO gene was found by whole exome sequencing of two Saudi Arabian siblings with autosomal recessive primary microcephaly [81]. However, this genetic variant was identified specifically only in this single consanguineous family.

#### 3.2.3. Neurodevelopmental Disorders

Other mutations on both AGMO alleles were reported in two unrelated children with a neurodevelopmental disorder (individual 1: p.Trp130Ter and p.Gly238Cys) (individual 2: p.Gly144Arg and p.Tyr236del) [82]. All four variants were devoid of enzyme activities as tested in a recombinant expression model in HEK293T cells. Individual 1 showed a delayed cognitive development, whereas individual 2 showed signs of autism spectrum disorders and had recurrent infections.

#### 3.2.4. Inflammation

Other reports proposed a role for AGMO in inflammation. From five Sudanese families (10 individuals, two of each family) with visceral leishmania (Kala-azar) relapses, rare mutations in the AGMO gene were found, of which variant 1 (c.701A>G, rs143439626, p.Lys234Arg) led to an amino acid exchange in exon 7 and variant 2 led to a stop in exon 12 (c.1213C>T, rs139309795, p.Arg405Ter) [83]. Both mutations were biochemically characterised, and results confirmed a loss of function only for mutation 2 (p.Arg405Ter), whereas the AGMO protein with mutation 1 (p.Lys234Arg) was proven to be functionally active in a recombinant expression model [84]. So far, it is not fully clear whether disrupted AGMO function alone or together with other mutated genes is causative for the recurrence of visceral leishmania and therefore needs to be further assessed.

#### 3.2.5. Heterotaxy

Rare copy number variations were found in TMEM195/AGMO associated in patients with heterotaxy [85]. In this study, sequencing data from 262 individuals with either complete *situs solitus* or complete *situs inversus* identified a deletion in exons 1–3 of 13 of the TMEM195/AGMO locus. In humans, this is the only evidence that AGMO could be involved in body axis formation during embryogenesis.

The distribution of AGMO variants on chromosome 7 around the AGMO locus is shown in Figure 3. Additionally, all mutations and variants including the corresponding reference are listed in Appendix A.

## 4. AGMO in Model Organisms

### 4.1. Mouse (Mus Musculus)

#### 4.1.1. AGMO in Macrophage Polarisation

Several studies have demonstrated differential regulation of AGMO expression in murine macrophages polarised to “M1” or “M2” types. Treatment of the murine macrophage-like cell line RAW264.7 with lipopolysaccharide and other pro-inflammatory agents led to a strong downregulation of AGMO [86]. Since these stimuli also lead to a strong production of platelet-activating factor (PAF), a role of AGMO in modulating the concentration of this powerful mediator has been suggested [86]. In addition, in primary murine bone-marrow derived macrophages, AGMO was strongly regulated by agents which alter macrophage polarisation [36]. Enzyme activities and expression levels were upregulated in alternatively activated M2 macrophages and strongly downregulated in classically activated M1 macrophages, which was achieved by treatment with lipopolysaccharide and interferon-gamma. Furthermore, the overexpression of *AGMO* in RAW264.7 cells had a profound effect on nitrite release and induced inducible nitric oxide synthase (iNOS/*Nos2*) expression upon pro-inflammatory stimuli, whereas knockdown had adverse effects and influenced the cellular lipidome by the accumulation of alkylglycerol species, which are the direct substrates for AGMO. Although a direct role for AGMO in PAF degradation has been suggested by overexpression in HEK293 cells [86], manipulation experiments in RAW264.7 cells did not influence PAF and lyso-PAF levels, possibly indicating a more complex regulation beyond the mere activity of AGMO [36]. Another study supported that *Agmo* expression was strongly suppressed by interferon-gamma and lipopolysaccharide in J774A1 macrophage-like cells and was increased in the M2 polarisation of murine adipose tissue macrophages treated with interleukin-4 for or with neuropeptide FF [87].

#### 4.1.2. AGMO in Experimental Colitis

AGMO was also reduced in the epithelium of the colon from mice in which colitis had been induced by dextran sodium sulphate [88]. Tetrahydrobiopterin treatment, the essential redox cofactor for AGMO, in dextran sodium sulphate-treated mice was able to reverse the pathological effects of provoked colitis. The authors hypothesised that this might be due to the modulation of AGMO activity as indicated by an altered lipid profile (i.e., clearance of 2-arachidonoylglycerol ether and lysophosphatidic acid species).

#### 4.1.3. Mouse Models Deficient in the AGMO Cofactor Tetrahydrobiopterin

In addition to nitric oxide synthases and aromatic amino acid hydroxylases, AGMO forms a third class of tetrahydrobiopterin-dependent enzyme systems [89]. Knock-out mice deficient in the first enzyme of tetrahydrobiopterin synthesis, guanosine triphosphate (GTP) cyclohydrolase 1, were reported to die early at embryonic day 13.5 due to bradycardia [90]. Mice deficient in the second enzyme of tetrahydrobiopterin biosynthesis, 6-pyruvoyl tetrahydropterin synthase, die after birth [91]. However, 6-pyruvoyl tetrahydropterin synthase knock-in mice with reduced tetrahydrobiopterin levels were viable and showed abnormal body fat distribution, elevated blood glucose levels, and alterations in lipid metabolism, which were assessed on a transcriptome level [92]. Other human and animal studies connected tetrahydrobiopterin to energy homeostasis and endothelial function in type 2 diabetes mostly by mechanisms of oxidative stress and nitric oxide synthase function [93,94,95,96]. From human genetic studies, a connection to AGMO function and energy metabolism in type 2 diabetes has been indicated and needs to be further analysed by functional analysis using cellular systems or model organisms for *in vivo* studies. AGMO knock-out mouse models are currently established in our laboratory in order to provide tools to study the role of AGMO in physiology in more detail.

### 4.2. The Nematode Caenorhabditis Elegans

#### 4.2.1. Importance of AGMO for Cuticle Stability

In *C. elegans*, mutations in genes required for tetrahydrobiopterin biosynthesis resulted in a loss of catecholamine and serotonin biosynthesis as expected, but the worms also showed a fragile cuticle. The occurrence of the fragile cuticle was independent of the impairment of catecholamine and serotonin biosynthesis, but this was observed to arise from the mutation of the gene encoding AGMO in this worm, which was termed *agmo-1* [97]. Loss of function mutations in tetrahydrobiopterin biosynthesis genes and in *agmo-1* were found in a genome-wide suppressor screen for resistance of the worm against Leucobacter *Verde-1* infections [21]. Thus, while causing a fragile cuticle, loss of function of the AGMO reaction makes the worm resistant to infections by Leucobacter *Verde-1*. Lipidomic analysis of the outer cuticle membrane demonstrated an accumulation of ether lipids and a more diverse lipid profile with longer acyl chain length in *agmo-1* mutants compared to wild-type nematodes. It was hypothesised that adaptations of the overall epidermal lipid composition of the worm are necessary to counteract the impact of accumulated ether lipids on membrane tension and bacterial attachment.

#### 4.2.2. Role of AGMO in Insulin-Like Signalling

Interestingly, genome-wide RNA interference screens for genes with effects on lifespan identified *agmo-1* (previous clone name BE10.2) as an important regulator for life span. Specifically, it was found to be associated with the *daf-2* insulin-like signalling pathway, since it had a greater impact on longevity in *daf-2* (orthologue of human insulin-like growth factor 1 receptor) mutants compared to wild type, but it had no effect in *daf-2 daf-16* double mutants. *Daf-16* is a transcription factor downstream of *daf-2* in the insulin-like signalling pathway in the worm [98]. The precise molecular mechanism of the effect of *agmo-1* in *C. elegans* ageing remains to be determined. From an overall perspective, functional AGMO plays a role in the host defence mechanism of *C. elegans* during bacterial infections by influencing the cuticle integrity via its lipid composition. Whether AGMO has a direct or indirect impact on the nematodes membrane integrity and might be able to modulate its constitution during infections or environmental stress is not known, since both studies were based on the absence of a functional protein.

### 4.3. The Clawed Frog Xenopus tropicalis

A role for AGMO in left–right axis patterning was proposed in patients with congenital heart disease [85]. To confirm this, morpholino oligomers were injected into *Xenopus tropicalis* oocytes to induce an RNA interference mediated knock-down of AGMO [99]. This resulted in disturbed cardiac looping, disrupted expression of laterality genes (*pitx2c* and *coco*), and delayed completion up to inhibition of gastrulation depending on the amount of morpholino injected. Furthermore, Wnt-signalling was disturbed and even dominant over β-catenin overexpression. Interestingly, a recent membrane protein two hybrid screen identified AGMO as a protein interactor of Wntless, which is a protein that regulates the secretion of Wnt signalling molecules from Wnt-producing cells [100].

### 4.4. Chicken (Gallus Gallus)

Vitamin E exerts various beneficial antioxidant properties and acts as a strong signal transducer. The effect of dietary supplementation with vitamin E isomers (α-, γ- tocopherol and a mixture of both) on liver and plasma gene expression profiles in chickens has been investigated by Affymetrix Microarrays [101]. Among 129 genes that were differentially expressed in all three treatment groups compared to the control group, AGMO was among the 10 top regulated candidate genes (8–10 fold upregulation upon either vitamin E isomer supplementation). In mouse tissues, AGMO is highly expressed and active in the liver [9], and this might be similar in chicken. Another study found a mutation in exon 8 of the AGMO locus (p.Ala312Thr), which was suggested to be a strong selection marker of Chinese gamecock chickens and was associated with their aggressive behavioural pattern [102]. The authors explained this on the basis of association studies, which linked rare copy number variations to neurodevelopmental disorders in humans [82,83].

## 5. Conclusions

Several biological associations and effects of AGMO have been collected over the past 10 years. Although the picture is very diverse, three complexes of evidence from different lines of research accumulate and call for further experimental research (Figure 4).

(i) AGMO and insulin-like growth factor signalling

Detailed and sound experimental evidence for a functional connection between AGMO and insulin-like growth factor signalling originates from genome-wide RNAi screens in the worm *C. elegans* [98]. Knockdown of AGMO shows one of the strongest effects of all genes on life span in *C. elegans*. Comparison of wild-type, *daf-2* mutants and *daf-2-daf-16* double mutants clearly suggests a role in the *C. elegans* homologue of insulin-like receptor signalling. This is complemented by several genome-wide association studies in humans that link a single nucleotide polymorphism in an intergenic region adjacent to the AGMO gene with fasting glucose levels in humans [40,41,42].

(ii) AGMO and Wnt signalling

Starting from observations of rare copy number variations to be associated with congenital heart disease in humans, unique genes in left–right patterning were identified [85]. Experimental studies in *Xenopus* demonstrated a similar behaviour of AGMO in organ development. The inhibition of AGMO by morpholino oligomers during *Xenopus tropicalis* embryogenesis implicated a requirement for AGMO to induce correct cardiac looping and to establish left–right asymmetry [99]. Furthermore, the overexpression of human AGMO in *Xenopus* embryos mimicked the effect of ectopic Wnt expression, which resulted in secondary axis formation. Biochemical studies with catalytically inactive forms of AGMO suggested that the enzymatic activity is necessary for the direct activation of beta-catenin-driven Wnt signalling. Intriguingly, experimental searches for protein binding partners for Wntless, which is a protein thought to be involved in the export of Wnt proteins from cells, yielded AGMO as a binding partner [100]. Whether or not associations of mutations in the AGMO gene with neurodevelopment and brain functions also is mediated by the Wnt pathway remains to be seen.

(iii) AGMO and infectious disease

Several studies in mouse macrophages consistently show a pronounced downregulation of AGMO upon treatment with pro-inflammatory stimuli and an upregulation of AGMO upon treatment with anti-inflammatory stimuli [36,86,87]. Although the mechanistic goal of these regulations is not yet understood, this supports speculations about a role of AGMO in innate immunity. This is complemented by genome-wide association studies in humans linking alterations in the AGMO locus with the outcome of leishmania infections [83] or with the course of tuberculosis [56].

## Figures and Tables

**Figure 1 life-11-00088-f001:**
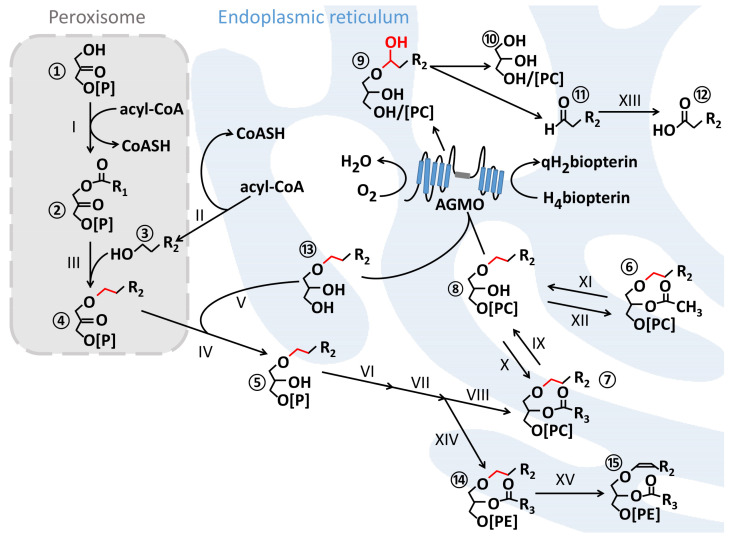
Ether lipid metabolism. Ether lipid synthesis is initiated in peroxisomes with the acylation of glycerone phosphate ① at the sn-1 position using acyl-CoA by glycerone-phosphate O-acyltransferase (I, GNPAT, EC 2.3.1.42) generating 1-acyl-glycerone-3-phosphate ②. Fatty acyl-CoA reductase (II, FAR1, EC 1.2.1.84) provides the fatty alcohol ③ for formation of the alkyl moiety by reducing acyl-CoA. Alkylglycerone phosphate synthase (III, AGPS, EC 2.5.1.26) introduces the alkyl group by exchanging the acyl side chain at sn-1 to produce 1-O-alkyl-glycerone-3-phosphate ④. The ether lipid precursor 1-O-alkyl-glycero-3-phosphate ⑤ can be either generated by acyl/alkyl-glycerone phosphate reductase (IV, EC 1.1.1.101) or from dietary alkylglycerols ⑬ by alkylglycerol kinase (V, EC 2.7.1.93). Further maturation to 1-O-alkyl-2-acyl-glycerophosphocholines ⑦ takes place at the endoplasmic reticulum in a sequential step by alkylglycerophosphate 2-O-acyltransferase (VI, EC 2.3.1.-), phosphatidate phosphatase (VII, EC 3.1.3.4) and diacylglycerol cholinephosphotransferase (VIII, EC 2.7.8.2). Alternatively, after reaction VII, ethanolamine-phosphotransferase (XIV, EC 2.7.8.1) attaches an ethanolamine—instead of the choline transferred by VIII—yielding plasmanylethanolamine ⑭, the substrate for plasmanylethanolamine desaturase (XV, EC 1.14.19.77), which is the enzyme that introduces the characteristic vinyl ether bond of plasmalogens ⑮. Alkylglycerol monooxygenase (AGMO) is located in the endoplasmic reticulum and is only able to cleave the ether bond (red) at the sn-1 of free alkylglycerols ⑬ or lyso-alkylglycerophospholipids ⑧ (here shown as cholines but also ethanolamines are accepted, however not the phosphatidic acid [6]) by using molecular oxygen and tetrahydrobiopterin (H_4_biopterin) as an essential co-factor. Lyso-alkylglycerophosphocholine ⑧ can be either generated from 1-O-alkyl-2-acyl-glycerophosphocholine ⑦ by phospholipase A2 (IX, EC 3.1.1.4) or from platelet-activating factor (⑥, PAF) by 1-O-alkyl-2-acetyl-glycerophosphocholine esterase (XI, EC 3.1.1.47). Both steps in lyso-alkylglycerophosphocholine ⑧ formation are reversible and use either 1-alkylglycerophosphocholine O-acyltransferase (X, EC 2.3.1.63) or 1-alkylglycerophosphocholine O-acetyltransferase (XII, EC 2.3.1.67). The AGMO enzymatic reaction creates a hemiacetal ⑨ that rearranges to the glycerol derivative ⑩ and a toxic fatty aldehyde ⑪ that is converted to the corresponding fatty acid ⑫ by fatty aldehyde dehydrogenase (XIII, FALDH, EC 1.2.1.48). R = carbon side chain; R1 = mostly 15 or 17 atoms (attached with an acyl-bond); R2 comprises 14 or 16 atoms (attached with an alkyl-bond); R3 contains at least 15 atoms with one or several double bonds (attached with an acyl-bond).

**Figure 2 life-11-00088-f002:**
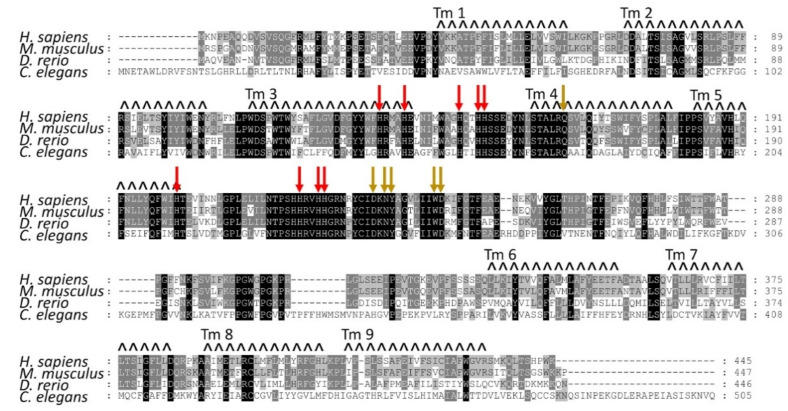
Sequence alignment of *Homo sapiens*, *Mus musculus*, *Danio rerio,* and *Caenorhabditis elegans* Agmo and AGMO topology. Red arrows indicate the nine conserved histidines at positions 132, 136, 145, 148, 149, 201, 221, 224, and 225 of the human AGMO, which shape the catalytic centre. Beige arrows show six additional residues that are all essential for AGMO activity and conserved across all species (Q162, D233, N235, Y236, W243 and D244 in the human AGMO sequence). The nine transmembrane domains (Tm) are marked with staggered lines (Tm 1, 40–61; Tm 2, 70–104; Tm 3, 111–137; Tm 4, 157–179; Tm 5, 183–201; Tm 6, 334–354; Tm 7, 363–383; Tm 8, 390–409; and Tm 9, 413–433 of the human AGMO sequence). Accession numbers for all species are as follows: NP_001004320.1 (*H. sapiens*), NP_848882.2 (*M. musculus*), NP_998048.2 (*D. rerio*), and NP_499664.2 (*C. elegans*). Program alignment of protein sequences was done by ClustalW incorporated into the MEGA-X package using default parameters and visualised by Genedoc (shading of similarity groups disabled).

**Figure 3 life-11-00088-f003:**
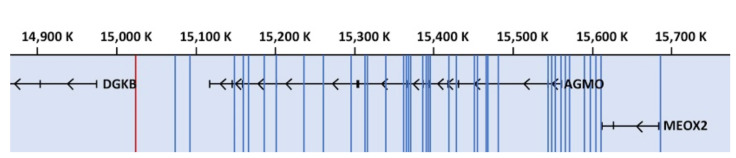
Schematic representation of AGMO variants implicated in human diseases. All known variations in the AGMO locus located on chromosome 7 (NC_000007.14) are shown schematically as blue lines. The red line in the intergenic region between diacylglycerol kinase beta (DGKB) and AGMO corresponds to single nucleotide polymorphism (SNP) rs2191349, which is associated with fasting glucose. Assembly GRCh38p12 (GCF_000001405.38).

**Figure 4 life-11-00088-f004:**
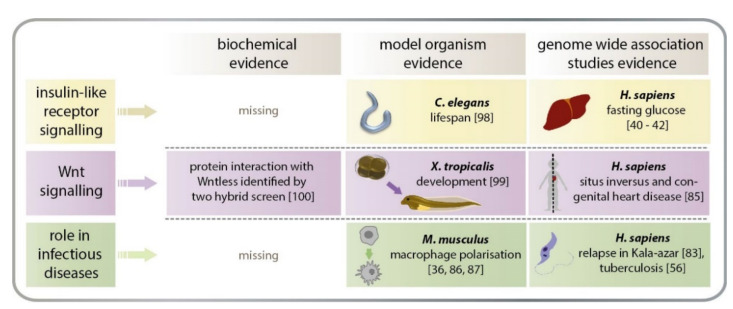
The emerging roles of AGMO. From human genetic association studies and experiments with model organisms, three plausible roles of AGMO can be expected. The involved pathways are (i) insulin-like signalling and longevity, (ii) Wnt signalling during early embryogenesis and axis formation, and (iii) infection control.

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
