# Peer review of "The Emerging Physiological Role of AGMO 10 Years after Its Gene Identification"

_life, 2021, doi:10.3390/life11020088_

Round 1

Reviewer 1 Report

Summary

The review summarizes the current knowledge about AGMO and its putative roles in health and disease gathered during the last 10 years after the identification of the gene encoding the AGMO protein.

General Comments

  • For an easier access into the review the section 3 (“alklylglycerols”) might be moved directly behind the introductory section

Minor Comments

  • A sequence alignment showing the conservation and important domains of the protein and/or the nucleotide sequence should be included. A comparison between the human, mouse, zebrafish and C. elegans sequence would be adequate.
  • A figure showing the AGMO reaction mechanism should be included, containing also the structures of all substrates of the enzyme known so far.
  • A table summarizing the known mutations and SNPs of AGMO known to be associated with human disease should be included to allow a better overview.

Author Response

We thank reviewer 1 for his detailed review and suggestions. We have now adapted the manuscript following the reviewer’s suggestions as it outlined below and are confident that the manuscript has greatly profited from this revision step.

  1. For an easier access into the review the section 3 (“alklylglycerols”) might be moved directly behind the introductory section.

We have moved the section on alkylglycerols right after the introduction as suggested.

  1. A sequence alignment showing the conservation and important domains of the protein and/or the nucleotide sequence should be included. A comparison between the human, mouse, zebrafish and C. elegans sequence would be adequate.

Please find the suggested alignment of AGMO protein of the four indicated species in the new figure 2. As reviewer 2 had a similar suggestion, to include also structural information on AGMO, we have combined the two requests by indicating those amino acid residues contained in one of the proposed nine transmembrane domains of AGMO and those involved in catalysis in the alignment.

  1. A figure showing the AGMO reaction mechanism should be included, containing also the structures of all substrates of the enzyme known so far.

Again, both reviewers have requested a similar figure and we have combined these requests to new figure 1 which includes ether lipid metabolism in peroxisomes and endoplasmic reticulum featuring the AGMO reaction in the middle and showing the AGMO substrates.

  1. A table summarizing the known mutations and SNPs of AGMO known to be associated with human disease should be included to allow a better overview.

We have added a table containing all known mutations and SNPs of AGMO as requested in supplementary Table S1.

Reviewer 2 Report

In this review article entitled “The Emerging Physiological Role of AGMO 10 Years After Its Gene Identification” describes the comprehensive overview of alkylglycerol monooxygenase (AGMO), which is the enzyme mediates the breakdown of ether lipids by catalyzing the cleavage of the ether bond. The paper covers human diseases related to AGMO, physiological studies with organismal models, and related knowledge about etherlipids. Though the review would be informative to gain more insights into this relatively new and unfamiliar protein, I suggest a few improvements to aid readers before publishing.

Major points:

Paragraphs are not properly separated throughout the manuscript so that the contents are not easy to understand (most sections consist of only single or two paragraphs with many contents). I suggest revise the paragraphing manner so that each paragraph contains one idea.

Because AGMO and ether lipids are not familiar to many readers, I suggest authors to include a metabolic map of ether lipids with indicating the reaction that AGMO mediates.

Similarly, “Introduction” and “AGMO in human diseases” sections are not very understandable and informative without graphical information. Please add a simple drawing or figure with which the readers can understand the structure of AGMO and the mutations/SNPs on AGMO gene described in “AGMO in human diseases” section.

Minor point:

In section2, the titles of the subsections are not consistent between subsections. I think it would be more kind if the authors provide animal names along with Latin names.

Author Response

We thank reviewer 2 for his detailed review. We have included all suggestions into the manuscript as is outlined below and feel that the manuscript has significantly improved due to this revision step.

  1. Paragraphs are not properly separated throughout the manuscript so that the contents are not easy to understand (most sections consist of only single or two paragraphs with many contents). I suggest revise the paragraphing manner so that each paragraph contains one idea.

We have followed the suggestion and have now revised the manuscript paragraphs in order to structure it more clearly and make the text easier readable and understandable.

  1. Because AGMO and ether lipids are not familiar to many readers, I suggest authors to include a metabolic map of ether lipids with indicating the reaction that AGMO mediates.

We have added new Figure 1 which includes ether lipid metabolism featuring the AGMO reaction in the centre.

  1. Similarly, “Introduction” and “AGMO in human diseases” sections are not very understandable and informative without graphical information. Please add a simple drawing or figure with which the readers can understand the structure of AGMO and the mutations/SNPs on AGMO gene described in “AGMO in human diseases” section.

We have prepared a new figure showing structural information on AGMO by combining it to the sequence alignment requested by reviewer 1. New Figure 2 now shows this alignment and includes information on the nine proposed transmembrane domains of AGMO as well as amino acid residues critical for catalysis.

Reviewer 1 requested a table for all AGMO mutations and the SNPs in the gene, which is now Table S1. As a graphical introduction to this table we have used the suggestion of reviewer 2 and have added a schematic representation of the AGMO gene and the location of the SNPs and mutations present in this locus (new Figure 3).

  1. In section 2, the titles of the subsections are not consistent between subsections. I think it would be more kind if the authors provide animal names along with Latin names.

We have adapted the titles of the subsections according to the reviewer’s suggestion.